# Electrolyte Design Strategies for Non-Aqueous High-Voltage Potassium-Based Batteries

**DOI:** 10.3390/molecules28020823

**Published:** 2023-01-13

**Authors:** Hong Tan, Xiuyi Lin

**Affiliations:** 1School of Materials Science and Engineering, Xihua University, 999 Jinzhou Road, Chengdu 610039, China; 2Key Laboratory for Biobased Materials and Energy of Ministry of Education, College of Materials and Energy, South China Agricultural University, 483 Wushan Road, Guangzhou 510642, China

**Keywords:** potassium-based batteries, high-voltage electrolytes, electrochemical window, cathode electrolyte interphase (CEI), ion solvation structures

## Abstract

High-voltage potassium-based batteries are promising alternatives for lithium-ion batteries as next-generation energy storage devices. The stability and reversibility of such systems depend largely on the properties of the corresponding electrolytes. This review first presents major challenges for high-voltage electrolytes, such as electrolyte decomposition, parasitic side reactions, and current collector corrosion. Then, the state-of-the-art modification strategies for traditional ester and ether-based organic electrolytes are scrutinized and discussed, including high concentration, localized high concentration/weakly solvating strategy, multi-ion strategy, and addition of high-voltage additives. Besides, research advances of other promising electrolyte systems, such as potassium-based ionic liquids and solid-state-electrolytes are also summarized. Finally, prospective future research directions are proposed to further enhance the oxidative stability and non-corrosiveness of electrolytes for high-voltage potassium batteries.

## 1. Introduction

Potassium-based batteries, including potassium-ion batteries (PIBs), potassium metal batteries (PMBs), and potassium-based dual ion batteries (P-DIBs), have gained steadily growing research input due to their vast potential for next-generation large-scale energy storage, as an alternative to lithium-based technologies [1,2,3]. The choice of potassium originates from its large abundance in the Earth’s crust and its environmental benignity as a natural recyclable element [4]. From the aspect of electrochemistry, the standard potential of the potassium-based redox couple, i.e., K^+^/K (−2.93 vs. SHE) is close to that of Li^+^/Li (−3.04 vs. SHE), suggesting a possibly high operation voltage and high energy density [5]. As shown in Figure 1a,b, despite its large atomic radius, K^+^ possesses the smallest and most flexible solvation structure among Li^+^ and Na^+^ in traditional carbonate electrolytes [6,7]. With its lower solvation energy, K^+^ also promises higher diffusivity and ionic conductivity, resulting from its fast and easier extraction from the solvation structures. Apart from the inherent benefits bestowed by the very nature of K, potassium-based batteries also hold prospects in terms of various available electrode materials and similar electrolyte systems with that of lithium-ion batteries (LIBs).

Intensive research on PIBs started in 2015, marked by the demonstration of graphite anode for electrochemically reversible K^+^ storage [8]. Subsequently, in the pursuit of high energy density (as stated by the formula: E (Wh/g) = Q (mAh/g) × voltage (V)), efforts were made towards optimization of electrodes and electrolytes [9,10]. On the negative side, metals with high capacity and appropriately low working voltages were investigated, such as bismuth (Bi), stibium (Sb), and potassium (K) [11,12,13]. On the positive side, high working plateaus over 4.0 V (vs. K^+^/K, as the standard reference hereafter unless else stated) were proven in polyanionic compounds such as KVPO_4_F and KVOPO_4_ [14,15]. Within the configuration of P-DIBs, an even higher working voltage of over 4.5 V could be achieved for graphite cathode, relying on the reversible intercalation of anions [16]. To match the electrodes, electrolytes with wide electrochemical windows (EWs) are highly demanded. As depicted in Figure 1c, the EW of a given electrolyte is determined by the energy difference (E_g_) between the lowest unoccupied molecular orbital (LUMO) and the highest occupied molecular orbital (HOMO). In an ideal situation, the thermodynamic stability of the system could be maintained with the electrochemical potential of the anode (*μ_a_*) below the LUMO and that of the cathode (*μ_c_*) above HOMO [17,18]. However, in practice, traditional carbonates usually suffer from restrained windows, leaving three major formidable challenges for electrolyte systems. First, the electrolyte will be oxidized and reduced on the cathode and anode side, respectively, for the first few cycles of a cell, along with the formation of passivation layers on both sides, namely, the solid electrolyte interphase (SEI) for the anode and the cathode electrolyte interphase (CEI) for the cathode, to re-gain thermodynamic equilibrium. Electrolyte decomposition could be suppressed if the passivation layer is stable and resilient. In addition to electrolyte decomposition, intense parasitic oxidation reactions take place at high potentials, resulting in continuous consumption of electrolytes and the release of gaseous contaminants such as O_2_, CO_2_, and CO [19]. The by-products of side reactions will impact the as-formed CEI, altering its component and thickness. The re-generation of effective CEI calls for successive electrolyte decomposition and interferes with the ongoing electrochemical reactions on the cathode side. Limited and declining Coulombic efficiency (CE) is thus seen. At the same time, CEI should be reasonably thin and uniform to withstand repeated expansion and shrinking of electrode materials and to allow fast and smooth ion migrations. Another issue concerning high voltage operation is current collector corrosion, since passivation layers often fail to protect the Al or stainless-steel current collectors over 4.5 V [20].

To break this predicament, the development of new potassium-based electrolyte systems suitable for high-voltage operation is imperative. Previous reviews have discussed the progress of electrolyte research for PIBs either from a broad view or focusing on a key point, such as effective SEI-formation-oriented electrolyte design, ester, and ether-based electrolyte design [3,6,9,10]. This review summarizes the research endeavors from the aspect of high-voltage applications, and two major study paths are introduced: (i) modify the traditional electrolyte systems via salt-solvent chemistry alteration, including concentration adjustment, the introduction of weakly solvating agents, applying multi-ion strategy and addition of additive; (ii) adopt new forms of electrolytes such as ionic liquids (ILs), polymer/gel-based solid-state-electrolytes (SSEs) systems. Meanwhile, the effect of CEI composition on high-voltage performance and the effectiveness to suppress Al corrosion by various methods are discussed.

## 2. Design Strategies for High-Voltage Organic Electrolytes

An electrolyte system consists of three key components, i.e., solvent, salt, and additive. The choice of solvent species sets its essential physical/chemical properties in the first place. Initial trials of solvents for PIBs begin with carbonates, as enlightened by their success in the commercialization of LIBs. The most studied carbonates include ethylene carbonate (EC), propylene carbonate (PC), diethyl carbonate (DEC), dimethyl carbonate (DMC), and ethyl methyl carbonate (EMC). Among them, EC and PC with cyclic structures are characteristic of high dielectric constant, which effectively inhibits electron migration into the electrolyte, as summarized in Table 1. By contrast, linear-type DEC, DMC, and EMC have low dielectric constant but low viscosity. 

For high-voltage operation, the thermodynamic stability of solvents is rather crucial. Figure 2 compares the LUMO and HOMO energy levels of different solvents [6]. As seen, EC shows more negative HOMO among carbonates, which predicts better anti-oxidation capability. When paired with a glassy carbon cathode, pure EC or PC has an oxidation limit of over 6.0 V [21]. When tested in a lithium system, it predicts an oxidation window of 5.5, 5.3, and 5.15 V (vs. Li^+^/Li) for EC, DMC, and DEC, separately [22]. However, EC-based electrolytes with a normal concentration of potassium salts, say, 0.8~1.0 mol/L (M), still face inevitable decomposition at high potentials in practice. This is particularly typical in the case of P-DIBs, where the intercalation/extraction of anions into/from graphite cathode takes place at over 5.0 V, resulting in limited CEs [23,24]. Table 2 summarizes the oxidation stability windows of typical normal-concentrated potassium-based electrolyte systems. As seen, the stability window ranges from 4.3~4.8 V for most ester-based systems. Moreover, compared with KFSI, KPF_6_ performs as a more promising salt for high-voltage operation, and more details will be discussed in Section 2.3. For ether-based electrolytes, the oxidation limit resides from 4.0~4.5 V. Taking account of the working potentials for high-voltage cathodes, which often exceed 4.7~4.8 V, current potassium-based organic electrolyte systems cannot meet the requirements. Therefore, several optimization strategies are proposed, such as adjustment of concentration, ion solvation structures and salt species, and addition of additives.

### 2.1. High Concentration

To push the limit, a high-concentration strategy is proposed. The solubility of the most used potassium salt KPF_6_ in carbonates is quite limited, with a saturated concentration of less than 1.0 M, stemming from the strong interaction between K^+^ and PF_6_^−^ and the consequent high solvation energy [6]. To reach a higher concentration, KFSI is selected as the salt due to the weaker interactive force between K^+^ and FSI^−^ and the lower dissociation energy of KFSI. A saturated 5.0 M KFSI-based electrolyte is then prepared using EC/DMC (1:1, *v*/*v*) as solvents, which proves an oxidation limit of 5.25 V [34]. When cycled with a graphite cathode in the range of 3.2–5.25 V, complete charge/discharge plateaus are evidenced indicating full FSI^−^ intercalation/de-intercalation (Figure 3a). By contrast, electrolytes with a concentration below 2.0 M cannot be charged over 5.0 V. Another merit of using high-concentration electrolytes in a dual-ion system is that it remarkably increases the cell-level energy density, since the electrolyte functions as an active material (Figure 3b).

The mechanism behind the expanded window with high concentration is then investigated. Anion–*x*K^+^ ion pairs existing in the electrolyte can be divided into three groups according to the number of surrounding ions: (i) solvent-separated ion pairs (SSIPs, *x* = 0), contact ion pairs (CIPs, *x* = 1) and aggregates (AGGs, *x* ≥ 2). Placke et al. probed the ion-coordination change of the KFSI-EMC electrolyte system with varied concentrations using Raman spectroscopy [25]. As shown in Figure 3c, the Raman band at 731 cm^−1^ representing FSI^−^ anion undergoes a continuous blue shift to 740 cm^−1^ when the concentration gradually changes from 0.5 M to 4.0 M, along with an obvious increase in intensity. Taking into account the estimated individual vibration frequency of SSIPs (730 cm^−1^), CIPs (737 cm^−1^), and AGGs (744 cm^−1^), it can be readily exposed that the dominated ion pairs change from SSIPs to CIPs, and ultimately to AGGs as salt concentration increases. For the solvent, the band at 931 cm^−1^ slowly collapses accompanied by the strengthening of the band at 939 cm^−1^, indicating apparent cation complexation with EMC molecules at high concentrations. The strong interactions of both FSI^−^–K^+^ ion pairs and K^+^–solvent complexes give rise to enhanced stability of such a system, enabling an oxidative limit up to 5.6 V (Figure 3d). A similar situation is also found in highly concentrated 4.0 M KFSI/EC-DMC electrolytes [26]. Researchers confirmed the strong interaction between K^+^ cations and FSI^−^ anions to form K^+^–FSI pairs and the aggregation of K^+^–FSI–EC–DMC complexes at high concentration, via analysis of ^17^O-nuclear magnetic resonance (NMR) spectra and Raman spectra (Figure 3e). As K^+^–FSI^−^–carbonate complexes form, the amount of free EC and DMC is evidently reduced (Figure 3f), making the electrolyte less susceptible to oxidation at high voltage. 

Another group of solvents showing promising properties is phosphates in the ester family. Except for their comparable wide EWs to those of carbonates (Figure 2), phosphates are bestowed with non-flammability for better safety. Tang et al. examined the solubility of KFSI in trimethyl phosphate (TMP) and found that it is able to reach a maximum concentration of 6.6 M [29]. At such high concentrations, nearly all the TMP molecules participate in solvating with K^+^ and FSI^−^ and minimal free ones remain. The oxidation ability of 6.6 M KFSI-TMP is as high as 5.4 V. Except for their enhanced anti-oxidation performance, phosphate-based concentrated electrolytes are inclined to present high affinity with both cathode and anode. Liu et al. discovered that triethyl phosphate (TEP)-based electrolyte with 2.6 M KFSI is capable of stabilizing the K_2_MnFe(CN)_6_ cathode as well as retaining high reversibility on the graphite anode, whereas 0.6 M KPF_6_/EC-DMC undermines the Coulombic efficiency of the K_2_MnFe(CN)_6_ cathode in the same cycling protocol [30]. 

Besides esters, ether-based solvents are also intensively studied. The most studied dimethyl ether (DME) is featured with low viscosity and the capability to facilitate the formation of thin, uniform, and robust SEI layers on the anodes. Such merit makes it an ideal solvent for graphite and K metal anodes in PIBs. Nonetheless, the charge limit voltages of ethers are fairly limited, as indicated by their high HOMO energy levels (Figure 2). Encouragingly, the high-concentration method also works for ether systems [35]. For example, the highly concentrated 7 mol/kg (m) KFSI/DME electrolyte demonstrates no apparent current flow upon anodic scan to 6.0 V by cyclic voltammetry (CV), whereas the 2 m one shows irreversible current flow above 3.5 V [36]. In addition, the Al current collector with 7 m KFSI/DME succeeds in maintaining its surface morphology after three cycles, which serves as corroborative proof for the wide EW of such concentrated electrolyte. Later on, concentrated diethylene glycol dimethyl ether (DEGDME)-based electrolyte is reported. Zhai et al. reveal the solvation structure change of K^+^ and FSI^−^ in DEGDME by means of Raman spectroscopy [32]. A similar evolution pattern is disclosed with that of KFSI in carbonates (Figure 4a). As the concentration of KFSI increases, more CIPs and AGGs arise with the K^+^ cation being the solvated core, instead, together with scarce free DEGDME left (Figure 4b). The intensified coordination of both K^+^–ether pairs and K^+^–anion–ether complexes are thus attributed to the enhanced oxidation limit. Moreover, FSI^−^ anions preferentially decompose upon the anodic process in concentrated ether-based systems, which promotes the formation of F-containing thin CEI with high ionic conductivity and shows prospects when matching with high-voltage cathodes [27].

### 2.2. Localized High Concentration/Weakly Solvating Strategy

Although high concentration brings improved thermodynamic stability, it also produces high viscosity and poor wettability. To overcome this dilemma, researchers constructed a localized high-concentration electrolyte (LHCE) by introducing co-solvents with low polarity. The 1,1,2,2-tetrafluoroethyl-2,2,2-trifluoroethyl ether (TFETFE), which is a highly fluorinated ether (HFE) with abundant -CF_2_ and -CF_3_ groups, is utilized as the functional co-solvent by Wu et al. [37]. The inclusion of proper-proportioned HFE endows the newly formed 2.76 m KFSI/DME-HFE electrolyte with high permeability and ionic conductivity. Interestingly, despite the decrease of the literal concentration, the as-formed LHCE reserves the K^+^/FSI^−^-centered CIPs and AGGs as uncovered in concentrated 6.91 m KFSI/DME, in partial areas, whilst HFE works more like a lubricant smoothing the channels between solvated groups and barely engages in solvating K^+^ or FSI^−^ (Figure 4c,d). Eventually, a high anodic limit of 5.3 V is achieved. Recently, Zhai et al. formed a new LHCE system by applying the same HFE as a co-solvent for a concentrated KFSI/DEGDME system [33]. Likewise, localized concentrated areas are present with all DEGDME molecules coordinating to K^+^ and FSI^−^, contributing to high voltage resistivity above 5.0 V. Free TFETFE molecules dilute the whole electrolyte system and help to guarantee high ionic conductivity. With 2.3 M KFSI in DEGDME/TFETFE, a 4 V-class PMB can be demonstrated. Given that HFEs hardly participate in the solvating reactions, they act as weakly solvating agents to tune the solution environments, making the LHCEs mildly solvating electrolytes. Such electrolyte systems are typical of low de-solvation energy for the cations, which is conducive to realizing smooth and fast cation storage on the anode side. In another respect, the weakened interaction between cations and solvent molecules allows reinforced cation-anion coordination, leading to an anion-induced type of SEI/CEI and enhanced anti-oxidation ability [38,39,40,41]. Via adopting an ether-based solvent, i.e., 1,2-diethoxyethane (DEE), Lu et al. constructed a weakly solvating electrolyte, the 1.0 M KFSI/DEE [31]. As shown in Figure 4e,g, through simulating the solvation structures of 1.0 M KFSI/DEE and 1.0 M KFSI/DME, researchers found that the K^+^-FSI^−^ clusters (K^+^(FSI^−^)*x*, *x* ≥ 2) overwhelmingly outnumber the K^+^-DEE structure (K^+^(FSI^−^)*x*, *x* = 0) and K^+^-FSI^−^ single pair (K^+^(FSI^−^)*x*, *x* = 2) in 1.0 M KFSI/DEE, with a ratio of 91.6%, 6.7%, and 1.7%, respectively. In comparison, the value obtained in 1.0 M KFSI/DME is 18%, 66%, and 16%, respectively, which is indicative of the close K^+^-solvent interaction but the weak connection between K^+^ and FSI^−^. The dominated K^+^-FSI^−^ clusters in 1.0 M KFSI/DEE help achieve better oxidative stability. As a result, the charge cut-off limit of a DEE-based electrolyte exceeds that of a DME-based one by 0.3~0.4 V under the same concentration.

### 2.3. Multi-Ion Strategy

The underlying mechanisms of the aforementioned methods reside in the re-arrangement of cation–anion and cation–anion-solvent solvation structures. Another approach to meet such a goal is combining different cations or anions by adding two or more salt species, scilicet the multi-ion strategy. By taking advantage of the merits of different components, the modified electrolytes demonstrate improved properties in various aspects, such as diffusivity, conductivity, reversibility, and electrochemical stability [42]. The “multi-ions” is usually related to the solvating processes by different ions. Typically, salts with different saturation concentrations could be used to expand the molarity upper limit to achieve a highly concentrated environment, especially in aqueous systems [43]. For example, the 1.0 M Zn(OAc)_2_ + 31.0 M KOAc, 21.0 M LiTFSI + 3.0 M ZnTfO_2_ and 20.0 M KCF_3_SO_3_ + 30.0 M KFSI hybrid formulae all produce the water-in-salt electrolytes (WISE), which assist in widening the EWs [44,45,46]. In non-aqueous systems, ion solvation structures could be readily altered by introducing a strongly solvating salt specie. Zhang et al. integrated 1.0 M NaClO_4_ in 0.5 M ZnOTf/TMP to dimmish free TMP molecules and form Na^+^/Zn^+^-based strongly solvated structures [47]. While extensively studied in lithium, sodium, and zinc-based batteries, the multi-ion concept is seldom implemented in potassium-based systems, partially as a consequence of the numbered optional potassium salts. In addition, this modification method is not fully understood, and more fundamental works to be addressed.

KFSI, KTFSI, and KPF_6_ are frequently used salts in organic solvents, with a decreasing solubility by sequence, whereas KClO_4_ and KBF_4_ hardly dissolve in carbonates. Due to the high dissociation energy, KPF_6_ only holds an utmost concentration of 0.9 m in PC. Conversely, KFSI with less de-solvation energy could reach a saturated concentration of 10 m in PC [36]. From another aspect, the study by Ming et al. revealed that the solubility difference between KPF_6_ and KFSI stems from the number of solvent molecules needed to solvate K^+^ ions [48]. As FSI^−^ shows higher coordinate capability with K^+^, fewer solvents will be necessarily demanded for the solvation process, thus leading to a significantly higher solubility of KFSI in carbonates and ethers. More importantly, the variation of solvating capability by two salts holds prospects once combining them in a single electrolyte system. Komaba et al. systematically studied the effect of *x*KPF_6_-*y*KFSI ratios on the performances of the KPF_6_/KFSI/EC/DEC electrolyte system [49]. The first finding states that with more KFSI in the formula, higher ionic conductivity is obtained, thanks to the intrinsic high ionic conductivity of KFSI (Figure 5a). Secondly, via tuning the mol ratio of KPF_6_/KFSI to 3 (0.75 KPF_6_–0.25 KFSI), the electrolyte with 1 m concentration demonstrates high oxidation ability and provides effective shielding on Al current collectors at 4.6 V (Figure 5b). The binary *x*KPF_6_-*y*KFSI formula is then investigated in the configuration of P-DIBs by Zhang et al. [28]. By fixing the total salt concentration at 0.8 M, different KPF_6_/KFSI ratios are tested with a ternary EC/DMC/EMC electrolyte system. Upon first anodic scan using a tungsten electrode, electrolyte with single anion KPF_6_ shows a high oxidation limit over 5.5 V, whilst that with pure KFSI displays an apparent decomposition current over 4.5 V. It is noteworthy that upon subsequent scans, KPF_6_-containing electrolytes exhibit a reinforced anti-oxidation capability. Further investigation reveals that it results from the formation of robust CEI with abundant F-containing species such as CHF-CH_2_ and CF_2_-CF_2_ species, which effectively passivate the cathode and prevent parasitic electrolyte oxidation owing to the high electron negativity of F element (Figure 5c). Combined with the high diffusivity of FSI^−^, the 0.8 M binary 0.5 KPF_6_–0.3 KFSI achieves a holistic electrochemical performance with graphite cathode when cycling from 3.0–5.5 V.

### 2.4. High-Voltage Additive

Introducing high-voltage additives into electrolytes is a tried-and-true method in LIBs towards the stabilization of cathodes with high cut-off voltages [51,52]. Additives with less negative HOMO than base solvents serve as sacrificing agents. They preferentially decompose at high potentials to form effective passivation layers, thus suppressing electrolyte oxidation and dissolution of transition metals from cathode materials. Given the numerous available studies in LIBs, additive research for PIBs is comparatively lagging. Among those, fluoroethylene carbonate (FEC) is a promising one. With an extra F atom bonding to the carbon atomic ring, FEC has lower HOMO than the original EC, implying its higher oxidation limit. Moreover, the inclusion of F within the solvent formulation facilitates the formation of F-containing species in CEI, which effectively restrain electrolyte degradation on the cathode side. Liu et al. found that the inclusion of FEC in EC/DMC can effectively stabilize the K_2_MnFe(CN)_6_ cathode at 4.5 V, along with an improved CE [30]. Via investigation of charge impedance on KVPO_4_F cathode with pure FEC, researchers found that the as-constructed CEI has a relatively high R_CEI_ value of ca. 1.5 kΩ/cm^2^, indicating high anti-oxidation ability [53]. By contrast, the R_ct_ representing the rest interfacial part other than CEI holds a value of 0.4 kΩ/cm^2^, which facilitates charge transfer through the interface.

However, FEC seems to impair the reversibility on the anode side [30]. Despite being frequently used in Li and Na-based systems, the effectiveness of FEC in K-based systems calls for further investigation. Via analyzing the compositions of FEC-induced SEI on hard carbon and potassium metal, it is found that insoluble KF and K_2_CO_3_ are dominating species in the passivation layers, which give rise to the increase in interfacial impedance on the anode side [54,55]. Apart from that, the existing KF is prone to defects, which makes the SEI less dense and susceptible to consecutive reduction. Considering the non-salt nature of additives, an excessive dosage might harm the overall electrolyte performance. Research by Yun et al. indicates that a key factor determining the efficacy of FEC is its concentration in the solution [56]. With a high fraction, say 5.0 wt%, the carbonate-based electrolyte tends to form SEI with densely distributed inorganic species, which induces high charge resistance. On the contrary, electrolyte with less FEC proportion, e.g., 0.2 wt%, produces thin SEI film with a proper proportion of inorganics such as KF, which enhances the reversibility of anode compared with that in the base carbonates. Therefore, with delicate control of FEC percentage, it could simultaneously stabilize the interface on both the cathode and anode sides.

Another potential high-voltage additive for PIBs is potassium nitrate (KNO_3_), as demonstrated by Kang et al. [50]. Regarding its less negative HOMO compared with that of KFSI and DME, KNO_3_ tends to decompose prior to KFSI and DME when charged to high voltage (Figure 5d). By adding 50 mM KNO_3_ into 2.3 M KFSI/DME, the upper EW expands from 3.9 V to 4.5 V (Figure 5e). Additionally, NO_3_^−^ will be preferentially adsorbed on Al current collector upon charging, as a result of the more negative adsorption energy of KNO_3_ than that of KFSI and DME, thus preventing Al corrosion by FSI^−^ anions (Figure 5f).

## 3. Adoption of Other Electrolyte Systems

Beyond traditional organic electrolytes, there are other promising alternatives for high-voltage applications, such as ILs and SSEs.

### 3.1. Ionic Liquids

ILs are liquids at room temperature composed of a pair of cation and anion-like salts. They usually feature low vapor pressure, high ionic conductivity, non-flammability, and wide EWs [57,58]. In particular, the exclusion of solvents ingeniously avoids obvious concentration gradients when both cations and anions are concurrently involved in electrode reactions. This feature finds its use in the prototype of P-DIBs in the first place. Winter et al. constructed a dual-graphite cell using N-butyl-N-methyl bis(trifluoromethanesulfonyl)imide (Pyr_14_TFSI) + 0.3 M KTFSI as electrolyte [59]. The as-built cell demonstrates a high rate to 5 C (1C = 50 mA/g) without capacity loss thanks to the high ionic conductivity of Pyr_14_TFSI/KTFSI electrolyte. Furthermore, with the addition of 2.0 wt% ethylene sulfite as an SEI-forming additive, the dual-ion cell exhibits ultra-stable cycling in the voltage range of 3.4–5.0 V for 1500 cycles, in conjunction with a high CE of over 98%. Another merit of ILs is that they can stabilize cathode materials at high potentials. Shikano et al. matched a high-voltage cathode K_2_Ni_1.75_Co_0.25_TeO_6_ with 1-propyl-1-methylpyrrolidinium TFSA (Pyr_13_TFSI) + 0.5 M KTFSI electrolyte [60]. When cycling within 2.7–4.5 V, it proves high reversibility in the structural evolution of K_2_Ni_1.75_Co_0.25_TeO_6_ and thus the corresponding electrochemical reactions. Moreover, an astonishing upper oxidative limit of over 6.0 V can be achieved for the Pyr_13_TFSI/KTFSI system. Besides dual-cation ILs, single K^+^ cation-based ILs are also proposed. The mixture of KFSI and potassium (fluor sulfonyl) (trifluoromethylsulfonyl)imide (KFTI) by a molar ratio of 55:45 (KFSI/KFTI) exhibits a melting point of 67 °C [61]. When scanned at 90 °C against a K metal counter electrode, the KFSI/KFTI electrolyte shows an oxidation voltage of 5.6 V. However, the drawback of such a system is evident, high operation temperature brings about additional safety concerns.

### 3.2. Solid-State Electrolyte

The development of SSEs is motivated by the demand for safer and more flexible cells, especially in the presence of metal-based electrodes, where the natural high mechanical strength of SSEs effectively suppresses the dendrite growth [62,63,64]. In this regard, several SSEs have been reported for PMBs, including gel polymer electrolytes (GPEs), composite gel polymer electrolytes (CGPEs), and crystalline organic electrolytes (COEs) [65,66,67]. For GPEs, the free-standing crosslinked polymer membrane containing salt species works as an electrolyte and separator at the same time. The properties of such an EES system depend largely on its ingredients. Liu et al. synthesized a CGPE using poly(vinylidene fluoride-hexafluoropropylene) (PVDF-HFP), KFSI, and polyacrylonitrile (PAN), where PAN functions as a membrane skeleton with reasonable strength and endows the CGPE with high-voltage capability thanks to its oxidation resistance [65]. As a result, the PVDF-HFP-KFSI@PAN has higher tensile strength and a wider EW than pristine PVDF-HFP-KFSI. Poly(methyl methacrylate) (PMMA) is another promising base polymer for high-voltage operation, owing to its high oxidation potential and a strong affinity with the absorbed organic liquids [66]. The as-formed PMMA-based GEP is proven to remain stable up to 4.9 V. Still, SSEs usually suffer from poor contact with an electrode on both sides due to their “solid” nature. In order to lower the interfacial resistance and keep a high CE, close contact between the electrode and SSEs is desired. Lee et al. designed a COE consisting of dimethyl sulfone (DMS) and KFSI [67]. By delicately adjusting the molar ratio of KFSI/DMS to 1/9, the as-prepared 1-KFSI/9-DMS shows anodic stability up to 5.8 V and a melting point below 94 °C. Intimate contact with the cathode is then realized by pouring the 100 °C-melt 1-KFSI/9-DMS onto high-voltage KVPO_4_F. When matched with K metal anode to form a KVPO_4_F/1-KFSI-9DMS/K solid state full cell, it demonstrates excellent cycling stability without capacity decay after 100 cycles within 2.5–5.0 V, thanks to the superb compatibility of 1-KFSI/9-DMS with both sides.

## 4. Summary and Perspectives

In summary, this review has discussed state-of-the-art modification methods for traditional organic electrolytes, ILs, and SSEs to expand their EWs for high-voltage potassium batteries. It is noteworthy that the requirements of high-performance electrolytes for either high-voltage potassium-based or lithium-based systems stay the same. That is, (i) high ionic conductivity to allow fast ion transfer; (ii) low electronic conductivity to hinder the migration of electrons; (iii) wide EWs; (iv) chemical stability over a normal operating temperature range, say 25 ± 20 °C; and (v) chemical stability in terms of formation of effective CEI and SEI [18]. For most studied traditional electrolytes based on esters and ethers, the mainstream way is to tune the ion-solvent interactions by controlling salt concentration, adding weakly solvating agent/co-solvent, and introducing different salt species with distinguished solvating capabilities. Enhancement of oxidative stability of such electrolytes is then realized through suppression of electrolyte decomposition, formation of effective CEI, and prevention of Al corrosion. Besides organic electrolytes, ILs and SSEs also hold prospects due to their intrinsic wide EWs and flexibilities towards safer and greener high-potential devices. Despite that, the research of high-voltage potassium batteries is in its nascent stage compared with high-voltage LIBs, where plenty of novel solvents and additives with remarkable anti-oxidative capabilities are available. The intrinsic difference between K and Li electrolyte systems originates from the variation of atomic sizes and Lewis acidity of K and Li, which bestows K systems with lower de-solvation energy and smaller volumes of solvated structure, thus higher ionic conductivity and diffusivity. While learning from the well-established knowledge from LIBs, a deep understanding of the very nature of K continues to be needed to design better high-voltage potassium-based electrolytes. Future explorations may focus on the following aspects:(1)Designing new high-voltage potassium salts. Due to the large size of K^+^ and its low Lewis acidity, the solubility of potassium salts is usually limited in most used carbonates. Even when utilizing high-concentration or multi-ion strategies, there are only a few choices stemming from that solubility limit, as summarized in Table 3. Therefore, adopting those with effective CEI-forming capabilities seems rather important. Lu et al. synthesized a new potassium salt based on cyclic anion hexafluoropropane-1,3-disulfonimide (HFDF^−^) [68]. By ingeniously incorporating three -CF_2_- groups in a single anion, the newly formed KHFDF facilitates the formation of thin, uniform, and F-abundant CEI together with a high oxidation resistance against Al up to 4.7 V (illustrated in Figure 6). This trial offers considerable inspiration for the development of high-voltage potassium salts.(2)Developing high-voltage solvents. Solvents are indispensable components setting the keynote of EWs for traditional organic liquid electrolytes. There are many high-voltage candidate solvents for LIBs, such as sulfone, nitrile, and fluorinated esters, which serve as significant references for the screening of proper solvents for potassium-based systems [69,70,71,72,73,74]. Among those, employing fluorinated solvents could be a promising way. One of the frequently fluorinated carbonates, FEC, possesses a higher anti-oxidative capability and is effective in forming robust F-abundant CEI when serving as the solvent for high-voltage PIBs. Moreover, the searching circle could be extended outside esters and ethers. For example, recently, a glyoxal-based electrolyte with an oxidative limit of 5.0 V has been reported for PIBs [75].(3)Modification of high-voltage battery components. Current collectors and cell cases are non-negligible factors affecting batteries’ reversibility, CE, and life when cycling in a wide range of EW. Some scientists have devoted themselves to understanding the formation mechanisms and compositions of passivation layers on typical current collectors, such as Cu, Al, and stainless steel [76]. Still, the dissolution and corrosion of current collectors cannot be fully eliminated. To ensure the long life of battery components, especially for stationary energy storage configurations, innovative modification methods for effective passivation are needed.

**Figure 6 molecules-28-00823-f006:**
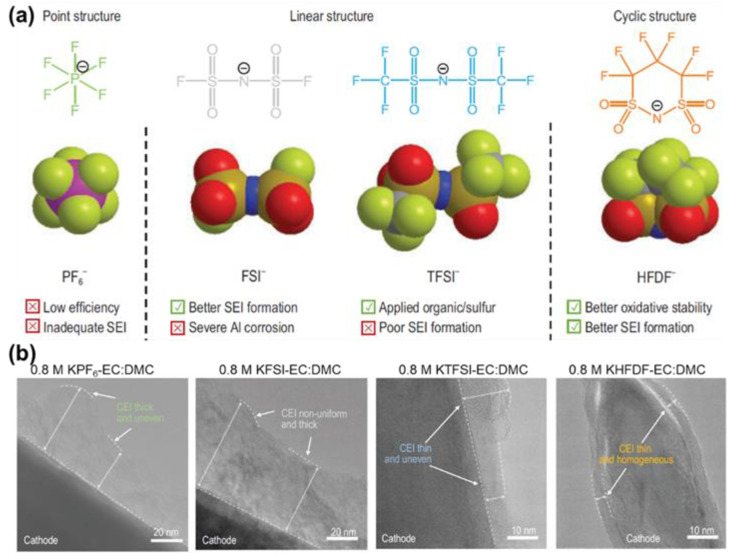
(**a**) Molecular structures and characteristics of PF_6_^−^, FSI^−^, TFSI^−^, and HFDF^−^ anions (Atoms in molecular structures: F−green; P−purple; N−blue; S−brown; O−red; C−grey). (**b**) Comparison of CEI with KPF_6_−, KFSI−, KTFSI−, and KHFDF−based electrolytes by TEM morphology. Reprinted with permission from Ref. [68]. Copyright 2022, Oxford University Press.

**Table 3 molecules-28-00823-t003:** Solubility (M (mol/L)/m (mol/kg) at 25 °C) of most used potassium salts in typical solvents.

Solubility	KPF_6_	KFSI	KTFSI	KClO_4_	KBF_4_
PC	0.9 m [36]	>10.0 m [36]		<0.5 M [36]	<0.5 M [36]
EMC	-	>4.0 M [25]		-	-
EC/PC	1.0 M [36]			-	-
EC/DEC	0.8 M [23]		>4.0 M [77]	-	-
EC/DMC	-	>5.0 M [34]		-	-
EC/DMC/EMC	0.7 M [28]	>5.0 M [28]		-	-
DME	0.9 m [36]	7.0 m [36]	6.0 m [36]	-	-
DEGDME		>4.0 M [32]		-	-
TMP	-	6.6 M [29]		-	-

## Figures and Tables

**Figure 1 molecules-28-00823-f001:**
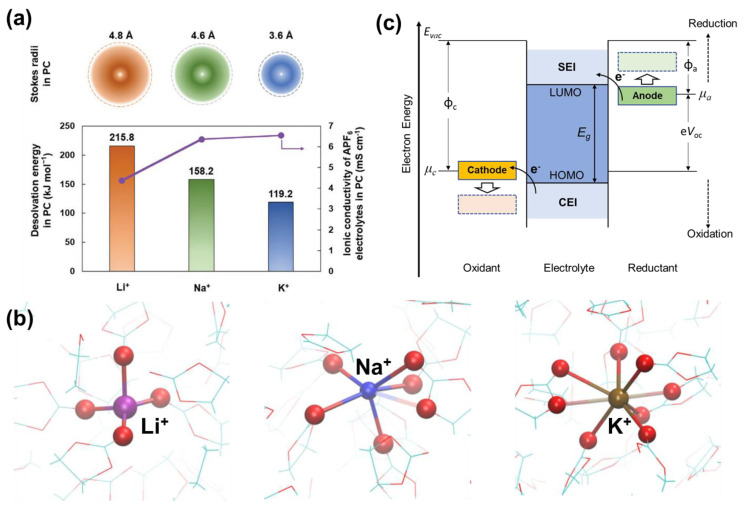
(**a**) Stokes radii, de−solvation energy, and ionic conductivity of APF_6_/PC electrolytes with different cations, i.e., A = Li^+^, Na^+^, K^+^. Reprinted with permission from Ref. [6]. Copyright 2021, John Wiley & Sons. (**b**) Typical solvation structures of Li^+^, Na^+^ and K^+^ in EC. Reprinted with permission from Ref. [7]. Copyright 2017, American Chemical Society. (**c**) Schemes of open−circuit energy diagram of PIBs (*μ_a_*, *μ_c_*: electrochemical potential of anode and cathode and relevant working function φ_a_, φ_c_; *E_g_*: electrochemical stability window; *E_vac_,* e*V*_oc_: vacuum energy level and open circuit voltage).

**Figure 2 molecules-28-00823-f002:**
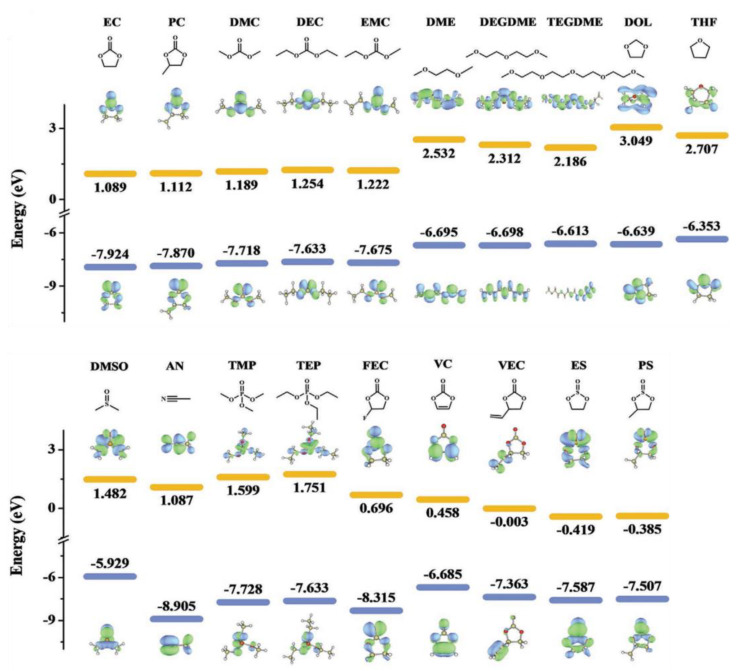
Molecular structures, LUMO (up), and HOMO (down) of state−of−the−art solvents and additives for PIBs. Reprinted with permission from Ref. [6]. Copyright 2021, John Wiley & Sons.

**Figure 3 molecules-28-00823-f003:**
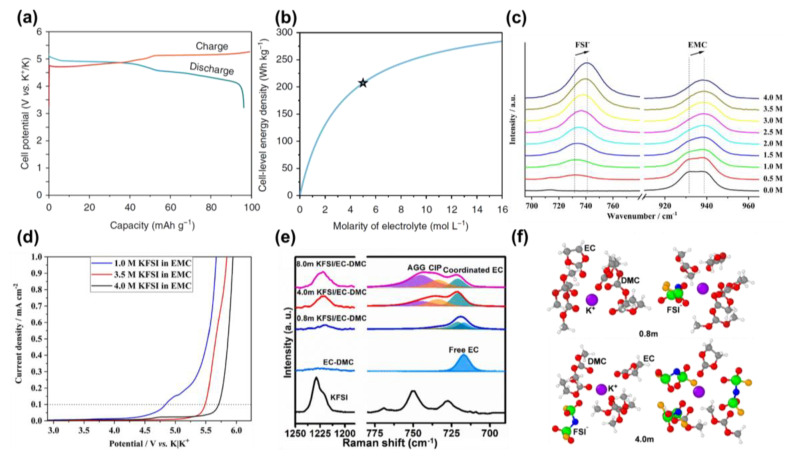
(**a**) Voltage profiles of KFSI−graphite P-DIB within 3.2–5.25 V; (**b**) Relationship between cell-level energy density and molarity of KFSI/EC-DMC electrolytes (The star represents the as-selected concentration, i.e., 5.0 M KFSI in EC/DMC (1:1, *v*/*v*)). Reprinted with permission from Ref. [34]. Copyright 2018, Springer Nature. (**c**) Normalized Raman spectra of KFSI/EMC electrolytes with a concentration of 0.5 M to 4.0 M; (**d**) Oxidative stability of KFSI/EMC electrolytes at different concentrations. Reprinted with permission from Ref. [25]. Copyright 2019, John Wiley & Sons. (**e**) Raman spectra of KFSI salt, EC−DMC solvent mixture, and KFSI/EC-DMC electrolytes by concentrations of 0.8 M, 4.0 M, and 8.0 M; (**f**) Solvation structures of K^+^ in KFSI/EC−DMC electrolytes with a concentration of 0.8 M (**up**) and 4.0 M (**below**) (Atoms in molecular structures: K−purple; S−green; N−blue; F−orange; O−red; C−grey; H−white.). Reprinted with permission from Ref. [26]. Copyright 2022, Elsevier.

**Figure 4 molecules-28-00823-f004:**
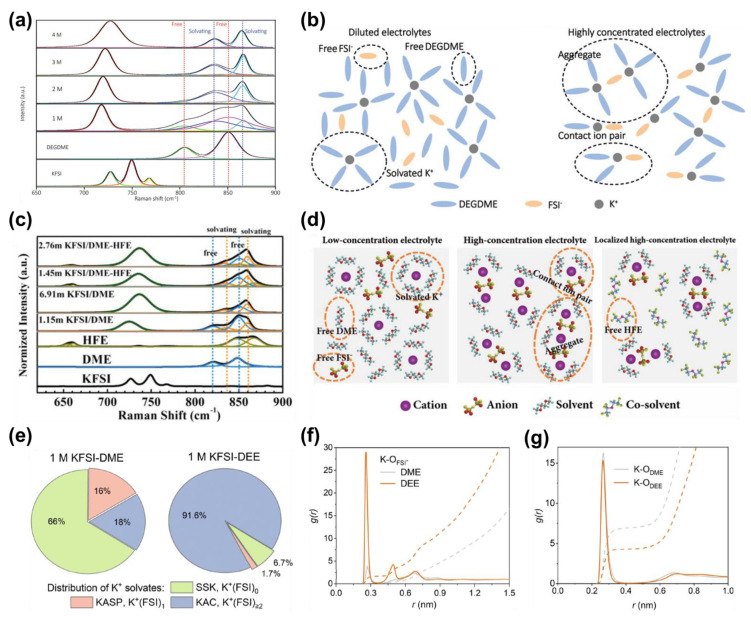
(**a**) Raman spectra of KFSI/DEGDME electrolytes with different concentrations (Fitting details: 805 cm^−1^ (light green line)/852 cm^−1^ (pink line) − gauche and trans conformation of C−C bond of DEGDME, respectively; 838 cm^−1^ (blue line) and 866 cm^−1^ (light blue line) – solvation of DEGDME molecules with K^+^ and FSI^–^) and corresponding (**b**) solvation structures. Reprinted with permission from Ref. [32]. Copyright 2021, Royal Society of Chemistry. (**c**) Raman spectra of KFSI, DME, HFE, 1.45/2.76 m KFSI/DME−HFE and 1.15/6.91 m KFSI/DME electrolytes (Fitting details: 820 and 850 cm^−1^ (blue line) – CH_2_ rocking and C−O stretching vibration for free DME molecules; 836 and 860 cm^−1^ (orange line) – solvation of DME molecules with K^+^ and FSI^–^); (**d**) comparison of solvation structures of low−concentration, high−concentration, and localized high−concentration electrolytes. Reprinted with permission from Ref. [37]. Copyright 2019, John Wiley & Sons. (**e**) Distributions of possible solvation compositions of 1.0 M KFSI/DME and 1.0 M KFSI/DEE by molecular dynamics simulations; radial distribution function (solid lines) and coordination number plots (dashed lines) of (**f**) K−O_FSI-_ and (**g**) K−O_DME_/K−O_DEE_. Reprinted with permission from Ref. [31]. Copyright 2022, John Wiley & Sons.

**Figure 5 molecules-28-00823-f005:**
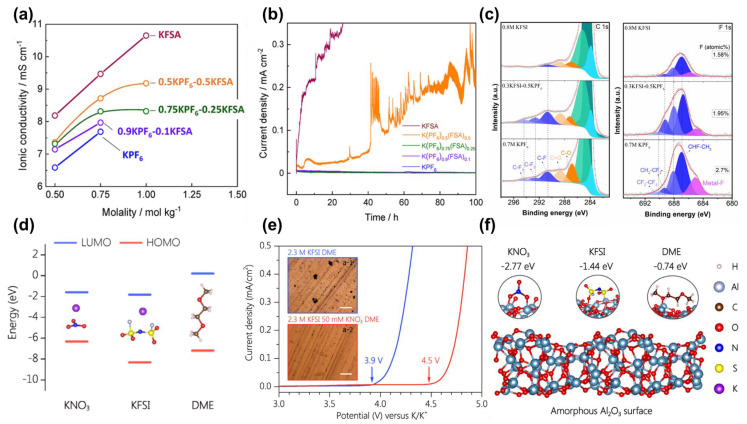
(**a**) Ionic conductivities of binary *x*KPF_6_−*y*KFSI/EC−DEC electrolytes with different KPF_6_*/*KFSI ratios and total molarities; (**b**) chronoamperograms of Al electrodes in *x*KPF_6_−*y*KFSI/EC−DEC electrolytes at 4.6 V. Reprinted with permission from Ref. [49]. Copyright 2020, American Chemical Society. (**c**) C 1 s and F 1 s surface XPS profiles of graphite electrodes cycled with 0.8 M KFSI, 0.5 KPF_6_−0.3 KFSI and 0.7 M KPF_6_. Reprinted with permission from Ref. [28]. Copyright 2021, Elsevier. (**d**) LUMO and HOMO of KNO_3_, KFSI and DME (Atoms in molecular structures: K−purple; N−blue; O−red; S−yellow; F−white; C−brown; H−pale pink); (**e**) LSV curves of Al electrodes in 2.3 M KFSI/DME electrolytes with/without KNO_3_ additive ( microscopic images of surface morphology of Al foil electrodes shown in the inset); (**f**) Comparison of adsorption energy of KNO_3_, KFSI, and DME molecules by Al_2_O_3_ surface. Reprinted with permission from Ref. [50]. Copyright 2021, Elsevier.

**Table 1 molecules-28-00823-t001:** Molecular structure, viscosity (η), dielectric constant (ε), melting point (T_m_), boiling point (T_b_), and low flash point (T_f_) of most used carbonate- and ether-based solvents for PIBs.

Solvent	Structure ^1^	η/cP (25 °C)	ε (25 °C)	T_m_/T_b_/T_f_ (°C)
EC	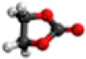	1.09 (40 °C)	89.78	36.4/248/160
PC	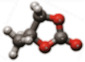	2.53	64.92	−48.8/242/132
DMC	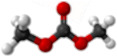	0.59	3.107	4.6/91/18
DEC	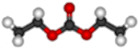	0.75	2.805	−74.3/126/31
EMC	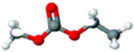	0.65	2.958	−53/110/-
DME	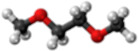	0.409	7.03	−58/85/−2

^1^ Atoms of molecular structures: carbon(C)—dark grey; hydrogen(H)—light grey/white; oxygen(O)—red.

**Table 2 molecules-28-00823-t002:** Summary of oxidation limits for normal-concentrated potassium-based electrolytes with different solvents.

Electrolytes	Examples	Working Electrodes	Oxidation Limit (vs. K^+^/K)	Ref.
Carbonates	1.0 mol/L KFSI in EMC	Pt	4.8 V	[25]
	0.8 mol/kg KFSI in EC-DMC (1:1 by vol.)	Al	4.5 V	[26]
	1.0 mol/L KFSI in EC-DEC (1:1 by vol.)	Al	4.6 V	[27]
	0.8 mol/L KFSI in EC-DMC-EMC (1:1:1 by wet.)	Tungsten (W)	4.5 V	[28]
	0.7 mol/L KPF_6_ in EC-DMC-EMC (1:1:1 by wet.)	W	5.5 V	[28]
Phosphates	0.8 mol/kg KFSI in TMP	Al	4.3 V	[29]
	1.0 mol/L KFSI in TEP	C-coated Al	4.4 V	[30]
Ethers	1.0 mol/L KFSI in DME	Al	4.5 V	[27]
	1.0 mol/L KFSI in DME	Al	4.0 V	[31]
	1.0 mol/L KFSI in DEGDME	Al	4.2 V	[32]
	1.0 mol/L KFSI in DEGDME	Al	4.0 V	[33]

## Data Availability

Not applicable.

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
