# Peer review of "Electrolyte Design Strategies for Non-Aqueous High-Voltage Potassium-Based Batteries"

_molecules, 2023, doi:10.3390/molecules28020823_

Round 1

Reviewer 1 Report

This manuscript presents a focused review of non-aqueous electrolyte design strategies for high-voltage K-based batteries. The authors first summarized modification methods for traditional organic electrolyte systems, such as concentration control, adoption of several salts and adding additives to enhance electrolytes oxidation stability, by means of adjustment of ion-solvent solvation structures. Then, ionic liquids and solid-state-electrolytes as potential alternatives for high-voltage usage were introduced. Finally, three perspectives were given in terms of design of new solvents, new salts and high-voltage-resistant battery components. Overall, this manuscript gives an in-time and relatively new review of the progress of high-voltage electrolytes for K-based batteries. It is recommended to publication after addressing some issues:

1. In section 2.3, the authors introduced “Multi-ion strategy” to achieved a holistic property of electrolyte by combining the merits of different salts. Is this enhancement relevant to the alternation of ion-solvent solvation structures?

2. CEI is critical in maintaining high-voltage stability. The authors claimed the inclusion of fluorine in CEI is significant for such goal. Besides that, are there any other features for the effective CEI possess?

3. In summary part, the authors commented that F-containing solvents are promising for high-voltage applications. Is it related to the intrinsic feature of such solvents or to their capability of forming F-abundant CEI?

Author Response

The authors deeply appreciate for the invaluable comments offered by the reviewer. All comments are now incorporated in the revision and the summary is presented in the following of how the amendments are made according to the individual comments (Please see the attachment).

  1. In section 2.3, the authors introduced “Multi-ion strategy” to achieved a holistic property of electrolyte by combining the merits of different salts. Is this enhancement relevant to the alternation of ion-solvent solvation structures?

Reply: The mechanisms behind the “Multi-ion strategy” is kind of complicated compared with that of high concentration and weakly solvating methods. It calls for extra research to fully understand this modification method. Still, we believe the enhanced performance by applying “multi-ions” is closely linked to the solvating processes by different ions. An evidence is the utilization of binary KPF6-KFSI salt systems. KFSI with less de-solvation energy takes priority in the insertion process towards graphite cathode while KPF6 with high solvation energy tends to immobilize electrolyte so as to reduce free solvent molecules and lead to high oxidative stability. Still, fundamentals need more works to be thoroughly understood.

The following sentences are added:

line 262: The “multi-ions” is usually related to the solvating processes by different ions.

line 274: In addition, this modification method is not fully understood and more fundamentals works to be addressed.

  1. CEI is critical in maintaining high-voltage stability. The authors claimed the inclusion of fluorine in CEI is significant for such goal. Besides that, are there any other features for the effective CEI possess?

Reply: Fluorine is positive element against oxidation at high potentials due to its high degree of electronegativity. However, the current research on CEI is relatively scarce compared with that on SEI. Taking account of the requirements that effective passivation layers should have on either anode or cathode side. We believe that besides fluorination nature, effective CEI is supposed to have a balanced composition including organic and inorganic species so as to maintain a moderate degree of elasticity and rigidness upon volume changes. Meanwhile, such CEI should be reasonably thin and uniform to withstand repeated expansion and shrink of electrode materials and to allow fast and smooth ion migrations.

The following sentences are added at Line 70:

“At the same time, CEI should be reasonably thin and uniform to withstand repeated expansion and shrink of electrode materials and to allow fast and smooth ion migrations.”

  1. In summary part, the authors commented that F-containing solvents are promising for high-voltage applications. Is it related to the intrinsic feature of such solvents or to their capability of forming F-abundant CEI?

Reply: We believe that fluorinated solvents possess higher anti-oxidative capability by containing F elements in their molecular structures. Meanwhile, the chances to form F-abundant CEI of such solvents upon decomposition should be higher than those without fluorination, like EMC and EC, thus F-containing solvents are promising for high-voltage applications.

The following sentences are modified at line 459:

“One of the frequently fluorinated carbonates, the FEC, possess higher anti-oxidative capability and effective in forming robust F-abundant CEI when serving as the solvent for high-voltage PIBs.”

Reviewer 2 Report

The design of electrolyte for post-lithium batteries is one of the important point for modern electrochemistry. I am sure that the review will be interesting for readers of the Molecules. But some improvement should be done

1. You should mention and analyse previous reviews about the electrolytes.

2. Explanation about differences between the previous reviews and your review should be done.

3. Advantages and perspectives the potassium-based batteries should be discussed in details. 

Author Response

The authors deeply appreciate for the invaluable comments offered by the reviewer. All comments are now incorporated in the revision and the summary is presented in the following of how the amendments are made according to the individual comments (Please see the attachment).

  1. You should mention and analyse previous reviews about the electrolytes.

Reply: As suggested, previous reviews about the electrolytes are summarized. The following sentences are added at line 76:

“Previous reviews have discussed the progress of electrolyte research for PIBs either from a broad view or focusing on a key point, such as effective SEI-formation-oriented electrolyte design, ester and ether-based electrolyte design [3,6,9,10]. This review summarizes the research endeavors from the aspect of high-voltage applications, and two major study paths are introduced”

  1. Explanation about differences between the previous reviews and your review should be done.

Reply: As suggested, the difference between the previous reviews and this review is given, as stated by the following sentence added at line 76:

“This review summarizes the research endeavors from the aspect of high-voltage applications, and two major study paths are introduced”

  1. Advantages and perspectives the potassium-based batteries should be discussed in details.

Reply: As suggested, a transition sentence is added between paragraph 1 and paragraph 2, to highlight the discussion about advantages of potassium-based battery technologies:

“Apart from the inherent benefits bestowed by the very nature of K, potassium-based batteries also holds prospects in terms of various available electrode materials and similar electrolyte systems with that of lithium-ion batteries (LIBs).”

More perspectives about potassium-based batteries are added in summary part at line 446:

“where plenty of novel solvents and additives with remarkable anti-oxidative capabilities are available. The intrinsic difference between K and Li electrolyte systems originates from the variation of atomic sizes and Lewis acidity of K and Li, which bestows K systems with lower de-solvation energy and smaller volumes of solvated structure, thus higher ionic conductivity and diffusivity. While learning from the well-established knowledge from LIBs, deep understandings of the very nature of K continues to be needed to design better high-voltage potassium-based electrolytes.”

Reviewer 3 Report

The author summarized the electrolyte design strategies, including the high concentration, localized high concentration, multi-ion strategy, and high voltage additive for non-aqueous high-voltage potassium-based batteries. Finally, the author provides perspectives for future research. The manuscript is well organized. The idea is clear and inspiring. I recommend the acceptance after minor revision.

1. As solubility is very important for electrolyte design, it would be better if the author could summarize the solubility of different K salts in different solvents.

2. Most of the electrolyte design strategy summarized in this paper is similar to that of the Li-ion battery. Could the author provide more comparison between electrolytes for K-ion battery and Li-ion battery? For example, what is the critical requirement of a K-ion battery other than a Li-ion battery?

Author Response

The authors deeply appreciate for the invaluable comments offered by the reviewer. All comments are now incorporated in the revision and the summary is presented in the following of how the amendments are made according to the individual comments (Please see the attachment).

  1. As solubility is very important for electrolyte design, it would be better if the author could summarize the solubility of different K salts in different solvents.

Reply: As suggested, the solubility of most used potassium salts in typical solvents are summarized in Table 3.

The corresponding sentence is added at line 465:

“ Even when utilizing high concentration or multi-ion strategies, there is only a few choices stemming from that solubility limit, as summarized in Table 3. Therefore, …….”

  1. Most of the electrolyte design strategy summarized in this paper is similar to that of the Li-ion battery. Could the author provide more comparison between electrolytes for K-ion battery and Li-ion battery? For example, what is the critical requirement of a K-ion battery other than a Li-ion battery?

Reply: More discussions about the requirements of electrolytes for both Li and K systems are incorporated at line 432 in summary part:

“It is noteworthy that the requirements of high-performance electrolytes for either high-voltage potassium-based or lithium-based systems stay the same. That is, i) high ionic conductivity to allow fast ion transfer; ii) low electronic conductivity to hinder migration of electron; iii) wide EWs; iv) chemical stability over normal operation temperature range, say 25 ± 20 ℃ and v) chemical stability in terms of formation of effective CEI and SEI [18].”

The merits of K system other than Li system is given by adding the following sentences at line 447 in summary part:

“The intrinsic difference between K and Li electrolyte systems originates from the variation of atomic sizes and Lewis acidity of K and Li, which bestows K systems with lower desolvation energy and smaller volumes of solvated structure, thus higher ionic conductivity and diffusivity. While learning from the well-established knowledge from LIBs, deep understandings of the very nature of K continues to be needed to design better high-voltage potassium-based electrolytes.”